# Safety and efficacy of allylamines in the treatment of cutaneous and mucocutaneous leishmaniasis: A systematic review

Jacob M. Bezemer[1,2]*, Jacob van der Ende[3], Jacqueline Limpens[4], Henry J. C. de Vries[5], Henk D. F. H. Schallig[1]

1 Experimental Parasitology Unit, Department of Medical Microbiology, Amsterdam UMC, University of Amsterdam, Amsterdam, The Netherlands, 2 Fundación Misión Cristiana de Salud, Shell, Pastaza, Ecuador, 3 Fundación Quina Care Ecuador, Puerto el Carmen de Putumayo, Sucumbíos, Ecuador, 4 Medical Library, Amsterdam UMC, University of Amsterdam, Amsterdam, The Netherlands, 5 Department of Dermatology, Amsterdam Institute for Infection and Immunity, Amsterdam UMC, University of Amsterdam, Amsterdam, The Netherlands

* j.m.bezemer@amsterdamumc.nl

**Data Availability Statement:** All relevant data are within the paper and its Supporting Information files.

## Abstract

Cutaneous and mucocutaneous leishmaniasis affect a million people yearly, leading to skin lesions and potentially disfiguring mucosal disease. Current treatments can have severe side effects. Allylamine drugs, like terbinafine, are safe, including during pregnancy. This review assesses efficacy and safety of allylamines for the treatment of cutaneous and mucocutaneous leishmaniasis. It followed the PRISMA statement for reporting and was preregistered in PROSPERO(CRD4201809068). MEDLINE, EMBASE, the Cochrane Central Register of Controlled Trials, the Global Health Library, Web of Science, Google Scholar, and clinical trial registers were searched from their creation to May 24th, 2020. All original human, animal, and *in vitro* studies concerning allylamines and cutaneous or mucocutaneous leishmaniasis were eligible for inclusion. Comparators—if any—included both placebo or alternative cutaneous or mucocutaneous leishmaniasis treatments. Complete cure, growth inhibition, or adverse events served as outcomes. The search identified 312 publications, of which 22 were included in this systematic review. There were one uncontrolled and two randomised controlled trials. The only well-designed randomised controlled trial that compared the treatment efficacy of oral terbinafine versus intramuscular meglumine antimoniate in 80 *Leismania tropica* infected patients showed a non-significant lower cure rate for terbinafine vs meglumine antimoniate (38% vs 53%). A meta-analysis could not be performed due to the small number of studies, their heterogeneity, and low quality. This systematic review shows that there is no evidence of efficacy of allylamine monotherapy against cutaneous and mucocutaneous leishmaniasis. Further trials of allylamines should be carefully considered as the outcomes of an adequately designed trial were disappointing and *in vitro* studies indicate minimal effective concentrations that are not achieved in the skin during standard doses. However, the *in vitro* synergistic effects of allylamines combined with triazole drugs warrant further exploration.

**Funding:** JB received a monthly volunteer allowance from Latin Link Nederland http://www.latinlink-nederland.nl/, which helped fund the study. JvdE is a volunteer for Fundacion Quina Care Ecuador and receives a monthly volunteer allowance from this organization which also helped fund the study. Latin Link and Fundacion Quina Care Ecuador had no role in study design, data collection and analysis, decision to publish, preparation of the manuscript. No additional external funding was received for this study.

**Competing interests:** JB is a volunteer for Latin Link Nederland and receives a monthly volunteer allowance from this organization. JvdE is a volunteer for Fundacion Quina Care Ecuador and receives a monthly volunteer allowance from this organization. There are no patents, products in development or marketed products associated with this research to declare. This does not alter our adherence to PLOS ONE policies on sharing data.'

## Introduction

Cutaneous leishmaniasis (CL) and mucocutaneous leishmaniasis (MCL), classified by the World Health Organization (WHO) as emerging neglected diseases, affect more than one million people yearly [1, 2]. CL manifests as skin lesions and MCL as potentially disfiguring mucosal disease of the nose, mouth, and larynx [3]. At least 20 different *Leishmania* parasite species can cause CL and MCL with differing clinical manifestations and responses to treatment [4]. Depending on the infecting *Leishmania* species, multiple treatment options are available but pentavalent antimonials (e.g., sodium stibogluconate and meglumine antimoniate) are still the most frequently used for American CL and MCL [5] and frequently used for old world leishmaniasis [6]. Yet, antimonial therapy is painful and requires multiple intralesional, intravenous, or intramuscular injections up to 30 days [5, 6]. Miltefosine, the oral alternative for systemic CL and MCL therapy, is not widely available and very expensive, limiting its use in clinical practice [7]. Moreover, pentavalent antimonials can result in hepatotoxicity, renal insufficiency, pancreatitis, cardiac arrest, and other serious side effects and there is no safe alternative systemic drug for use in pregnant women [8, 9]. Furthermore, depending on the region and species, poor treatment responses exist for pentavalent antimonials and miltefosine [10]. Consequently, there is a pressing need to identify alternative oral, safe, available, affordable, and efficacious treatment options for CL and MCL.

Thirty years ago, Goad et al, reported an inhibitory effect of terbinafine on cultured promastigotes of the *Leishmania mexicana* complex species [11]. Terbinafine is a member of the allylamine drug group, together with butenafine and naftifine.

Allylamines inhibit squalene-2,3-epoxidase causing accumulation of squalene and depletion of sterols in *Leishmania* amastigotes, resulting in growth inhibition and parasite death [12]. Terbinafine is used as a first line oral treatment for fungal infections and is the preferred systemic treatment for toenail infections in elderly people for safety reasons. Because of its use as antifungal, terbinafine is widely available in pharmacies all over the world at reasonable prices in oral and topical formulations [13].Terbinafine might be a safe systemic option in pregnancy, as no teratogenic side effects have been described [14, 15].

Since allylamines might be an attractive alternative CL and MCL treatment option, a systematic literature review was performed to assess the efficacy and safety of allylamines in CL and MCL treatment and to define priorities for future investigations. All original human, animal, and *in vitro* studies concerning allylamines and CL or MCL were eligible for inclusion. Comparators—if any—included both placebo or alternative CL and ML treatments. Cure rate in humans, change in lesion diameter in animals, promastigote and amastigote viability and growth, and adverse events served as outcome.

## Methods

### Search strategy

This systematic review, registered in PROSPERO (registration number CRD42018090687, 2018) and available at: https://www.crd.york.ac.uk/prospero/display_record.php?RecordID=90687, followed the Preferred Reporting Items for Systematic Reviews and Meta-analyses (PRISMA) guidelines [16].

A medical information specialist (JL) searched the following electronic databases for studies on leishmaniasis and allylamines, using controlled terms and text words, from their creation to May 24th, 2020: MEDLINE (OVID), EMBASE (OVID), the Cochrane Central Register of Controlled Trials (CENTRAL), The Global Health Library, Web of Science, Google Scholar (1st 150 hits) and the clinical trial registers, ClinicalTrials.gov and WHO_ICTRP. No language,

date or other restrictions were applied. The complete search strategies are presented in the S1 File. Reference lists and the citing articles of the identified relevant papers were cross-checked in Web of Science for additional relevant studies. The records retrieved were imported and de-duplicated in EndNote.

## Study eligibility

All original human, animal, and *in vitro* studies were eligible if they examined the effects of systemic or topical allylamines with the following endpoints: cure rate in humans, skin lesion diameter in animals and promastigote or amastigote *in vitro* growth or viability in the laboratory. The presence of *Leishmania* parasites had to be confirmed in the study by either microscopy, culture, or molecular techniques. If one *Leishmania* species was known to cause >90% of the CL or ML cases in the study area in human studies, this species might be assumed as the causative species in all patients.

## Study selection

JB and JvdE independently screened the identified studies using EndNote and resolved differences through discussion or consultation with a third reviewer (HS). Studies included during title and abstract screening were subsequently assessed as full text. Authors of conference abstracts were contacted to request unpublished data. If the full report was written in another language than English, Spanish, German, Dutch, French, or Portuguese authors were requested to provide a translation.

## Data extraction

The following data from all included studies were entered in Excel: study setting, study population, probable *Leishmania* species, allylamine studied and treatment combinations. From human studies the following information was recorded: age, gender, lesion type and duration, drug presentation, treatment scheme, cure rates, adverse events, and information for assessment of risk of bias. Cure rates were calculated according to intention to treat analysis and cure was defined as complete epithelialization of ulcers or decrease in induration size > 75% of nodules at last available follow up.

From animal studies the following information was recorded: age, gender, lesion type and duration, drug presentation, treatment scheme, effect on lesion diameter, adverse events, and information for assessment of risk of bias. From *in vitro* studies the following information was recorded: drug concentrations, promastigote or amastigote growth or viability, and culture cytotoxicity. JB and JvdE extracted data in duplicate and resolved differences through discussion.

## Risk of bias assessment

JB and JvdE independently assessed the quality of the clinical trials and resolved differences through discussion. Randomised controlled trials were assessed using the revised Cochrane collaborations tool (RoB2) and non-randomised controlled trials with the Cochrane tool for non-randomised controlled trials (ROBINS-1) [17, 18]. Animal studies were assessed with the SYRCLE´s risk of bias assessment tool [19]. *In vitro* studies were assessed with the tool developed by the United States national toxicology program [20]. Results of risk of bias assessments were visualized using the Cochrane risk-of-bias visualization tool [17].

# Results

## Studies included

The literature search identified 312 manuscripts of which 75 were included for full text assessment after screening of titles and abstract. After full text examination, 22 studies were included. Major reasons for exclusion were: 'different topic´ and 'textbook or review´. The data of two conference abstracts, could not be retrieved by contacting the authors [21, 22], and were therefore excluded from the study. The authors of a study presented in Chinese and another study in Farsi could not provide the data or the English translation of the report and these studies were therefore excluded [23, 24] (Fig 1).

Characteristics of the included human trials (n = 3) [25–27], mice studies (n = 2) [28, 29], and amastigote and promastigote studies (n = 12) [11, 30–40] are presented in Tables 1 and 2. The case reports (n = 5) [41–45] are presented in S1 Table.

## Risk of bias assessment

Two randomised controlled clinical trials [25, 26], a one arm non randomised trial [27], and two animal trials [28, 29] were assessed for risk of bias. Farajzadehs randomised controlled trial in 2015 had an acceptable risk of bias [25]. The study of Farajzadeh from 2016 lost 73% of patients to follow up [26] and Bahamdans study had severe deviations from intended interventions and 48% loss to follow up [27], leading to an overall judgement of high risk of bias for both. The two mice studies suffered from high risk of bias in various domains including allocation concealment and blinding of outcome assessment [28, 29]. The 12 *in vitro* studies presented minor methodological risks of bias (Figs 2–5).

It was not possible to perform a meta-analysis of the study outcomes, due to the small number of studies, their heterogeneity and low quality.

## Efficacy of terbinafine on *L. tropica* human infections and adverse events

The clinical trial of Farajzadeh 2015 [25] included 80 *L. tropica* infected patients randomised between two different treatment groups: 1) oral terbinafine 125-500mg (weight dependent) daily during four weeks, combined with cryotherapy every two weeks (n = 40) and 2) meglumine antimoniate 15mg/kg/day for three weeks combined with cryotherapy every two weeks (n = 40). Complete follow up was achieved for all patients at three months. Contrary to the hazard ratios presented by Farajzadeh, in this review the endpoint was the complete cure rate. In the terbinafine arm 15/40 (38%) patients were cured and in the meglumine antimoniate arm 21/40 (53%) cases were cured, a difference that was not statistically significant in Kaplan Meier analysis (*p* = 0,39). None of the *in vivo* studies reported adverse events [25–29].

## Species specific effectivity of allylamine treatments

Growth inhibition of terbinafine in promastigote *in vitro* studies was reported for the *L. major*, *L. tropica*, *L. mexicana*, *L. braziliensis*, *and L. guyanensis* complexes. An interspecies comparison in promastigote cultures with terbinafine 27µM showed higher inhibition levels in old world (*L. major*, *L. tropica*, and *L. aethiopica*) species (Table 3).

Treatment with butenafine killed *in vitro* cultured amastigotes of *L. amazonensis* and *L. braziliensis* at a mean effective dose of 30–38µM compared to the median cytotoxic concentration of 98µM [35]. Naftifine killed *L. major* amastigotes with mean effective dose of 45µM whilst cytotoxicity levels were more than 110µM [34].

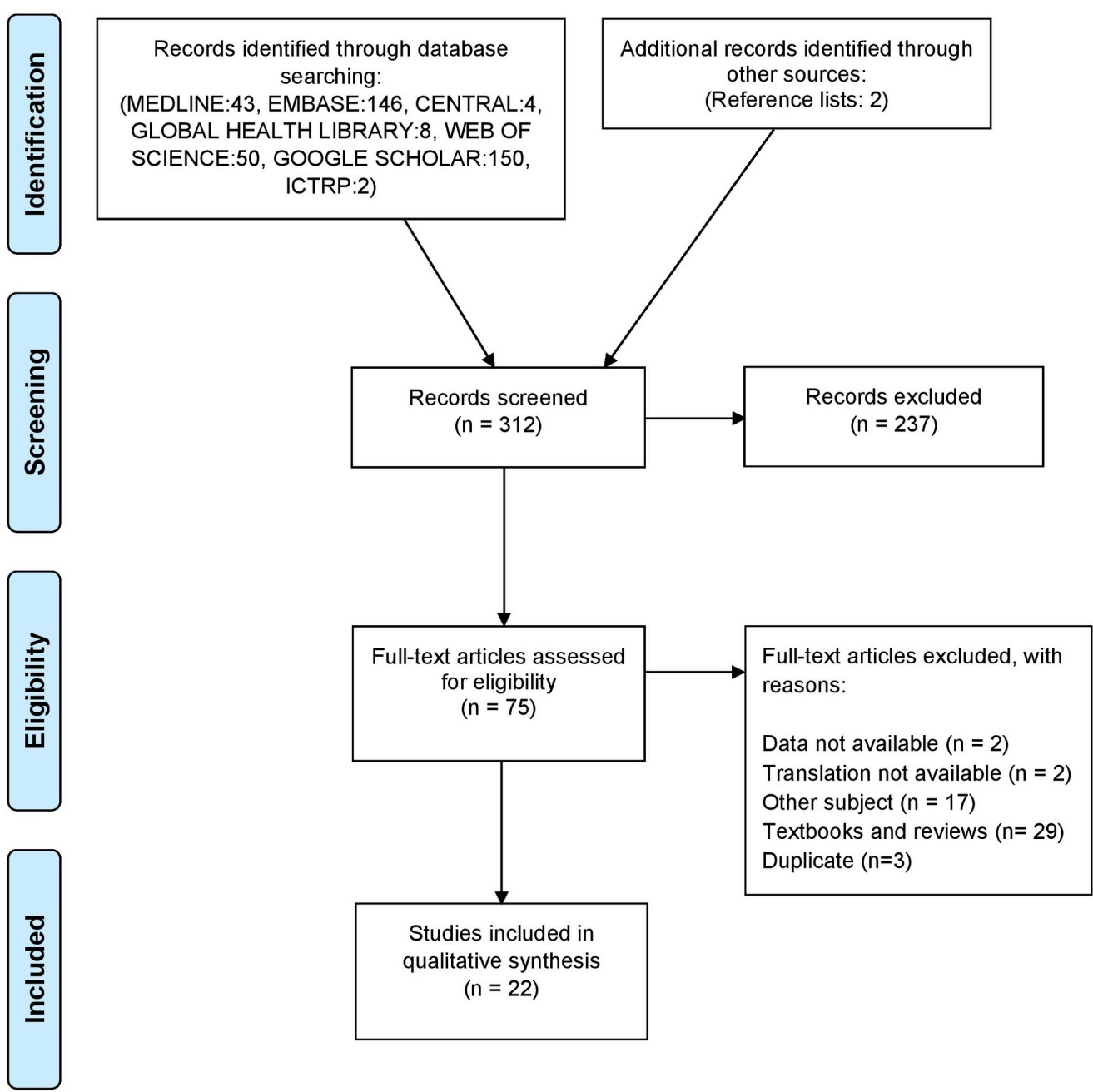

**Fig 1. PRISMA literature assessment flow diagram.**

## Case reports reporting cure with terbinafine

There were five case reports showing a curative effect of terbinafine. In two case reports terbinafine cured a *L. tropica* infected patient although the reason to start terbinafine was unclear [41, 42]. An HIV positive patient infected in Colombia and initially diagnosed with a skin mycosis, was treated, and cured with terbinafine when CL was diagnosed eventually. The causative *Leishmania* species was unknown [43]. In another case, terbinafine 250mg daily combined with a Crotamiton 10% + Sulphur 2% cream in the absence of other CL treatments cured a Kenyan patient with CL; the causative *Leishmania* species was unknown [44].

**Table 1. Characteristics of human and mice trials reporting treatment of cutaneous leishmaniasis with terbinafine.**

| Zakai [28] | Sampaio [29] | Farajzadeh [25] | Farajzadeh [26] | Bahamdan [27] | First Author |
|---|---|---|---|---|---|
| 2000 | 2003 | 2015 | 2016 | 1997 | **Year** |
| *major* | *amazonensis* | *tropica* | *tropica* | *tropica* | *Leishmania* **species** |
| 20 | 15 | 40 | 44 | 27 | **Number of participants** |
| 40 | 29 | 40 | 44 | NA | **Number of controls** |
| systemic | systemic | systemic | topical | systemic | **Presentation** |
| NA | NA | cryotherapy | meglumine antimoniate | NA | **Combination** |
| 0,2mg | 100mg/kg | 125-500mg | 32,25–75.5mg | 500mg | **Dose / day** |
| 28 | 20 | 28 | 20 | 28 | **Days treated** |
| untreated | placebo | meglumine antimoniate + cryotherapy | meglumine antimoniate + placebo | NA | **Control 1 treatment** |
| Itraconazole | sodium stibogluconate | NA | NA | NA | **Control 2 treatment** |
| 5[c] | 35[b] | NA | NA | NA | **Mean Lesion diameter (mm)** |
| 7 | 36 | NA | NA | NA | **Control 1 mean lesion diameter** |
| 1[c] | 28[c] | NA | NA | NA | **Control 2 mean lesion diameter** |
| NA | NA | 0,38 | 0,14 | 0,15 | **Cure rate[a]** |
| NA | NA | 0,53 | 0,20 | NA | **Control cure rate[a]** |
| none | none | none | none | none | **Adverse event rate** |

NA: Not Applicable

[a] Defined as complete epithelialization of ulcers or decrease in induration size > 75% of nodules at last available follow up and calculated according to intention to treat analysis

[b] no significant difference with untreated controls

[c] significant difference with untreated controls

Terbinafine 500mg combined with itraconazole 200mg daily for six months was started without evident reason in a patient suffering from MCL, visceral leishmaniasis, and liver cirrhosis caused by *L. infantum*. Terbinafine proved surprisingly effective resulting in the cure of the nasal mucosal inflammation and improvement of the liver function [45].

## Terbinafine drug combination treatment

Various *in vitro* studies evaluated the combination of terbinafine with drugs from the triazole group. Up to 300-fold improvement was demonstrated of the inhibition of *L. braziliensis* promastigotes when combining ketoconazole with terbinafine [37]. Another study reported that ketoconazole and terbinafine had a synergistic effect on the inhibition of *L. amazonensis* amastigotes resulting in a minimally inhibitory concentration of 0,001μM (Table 4) [39].

## Discussion

This systematic review assesses efficacy and safety of allylamines for the treatment of CL and MCL. It comprises an exhaustive search of eight electronic databases and trial registers. It assesses the risk of bias of two randomised controlled trials, a non-controlled trial, two animal studies, and 12 *in vitro* studies and summarizes the available evidence including five case reports. Generally, the quality of evidence was low and human studies were done only in *L. tropica*.

**Table 2. Characteristics of *in vitro* studies reporting on effects of allylamines in cutaneous and mucocutaneous leishmania species.**

| First Author | Year | *Leishmania* species | Allylamine | Combination | *Leishmania* model | Amastigote host cell type | Host cell toxic concentration (μM) | Effective concentration (μM) | Parameter of effectivity |
|---|---|---|---|---|---|---|---|---|---|
| Andrade Neto [30] | 2013 | *amazonensis* | terbinafine | LBqT01 / imipramine | amastigote | mice peritoneal macrophages | 80 | 23 | IC50 |
| Andrade Neto [30] |  |  |  | imipramine | promastigote | NA | NA | 15 | IC50 |
| Andrade Neto [31] | 2011 | *amazonensis* | terbinafine | NA | promastigote | NA | NA | 8 | IC50 |
| Andrade Neto [32] | 2009 | *amazonensis* | terbinafine | NA | promastigote | NA | NA | 4–9 | IC50 |
| Beach [33] | 1989 | multiple | terbinafine | NA | promastigote | NA | NA | 27 | 26–93% growth inhibition |
| Berman [34] | 1987 | *major* | terbinafine | NA | amastigote | human monocyte derived macrophages | >110 | 31 | ED50 |
| Berman [34] |  |  | naftifine | NA | amastigote | human monocyte derived macrophages | >110 | 45 | ED50 |
| Bezerra Souza [35] | 2016 | *amazonensis / braziliensis* | butenafine | NA | amastigote | mice peritoneal macrophages | CC50: 98 | 30–38 | ED50 |
| Bezerra Souza [35] |  |  |  | NA | promastigote | NA | NA | 34–81 | ED50 |
| Chance [36] | 1999 | *amazonensis* | terbinafine | NA | promastigote | NA | NA | 34 | >2-fold increase of squalene |
| Goad [11] | 1985 | *mexicana* | terbinafine | NA | promastigote | NA | NA | 34 | MIC |
| Rangel [37] | 1996 | *mexicana / braziliensis* | terbinafine | ketoconazole / D0870 | promastigote | NA | NA | 5–15 | MIC |
| Tariq [38] | 1994 | *tropica* | terbinafine | NA | promastigote | NA | NA | 1373 | MIC |
| Vannier-Santos [39] | 1995 | *amazonensis* | terbinafine | ketoconazole | amastigote | mice peritoneal macrophages | NA | 0,001 [a] | MIC |
| Vannier-Santos [39] |  |  | terbinafine | ketoconazole | promastigote | NA | NA | 1 | MIC |
| Zakai [40] | 2003 | *major* | terbinafine | NA | promastigote | NA | NA | 6 | IC50 |
| Zakai [40] |  | *mexicana* | terbinafine | NA | promastigote | NA | NA | No effect | NA |

N.A: Not Applicable, IC50: Half maximal inhibitory concentration, ED50: Median effective dose, MIC: Minimum inhibitory concentration

[a] combined with 0,001 μM Ketoconazole

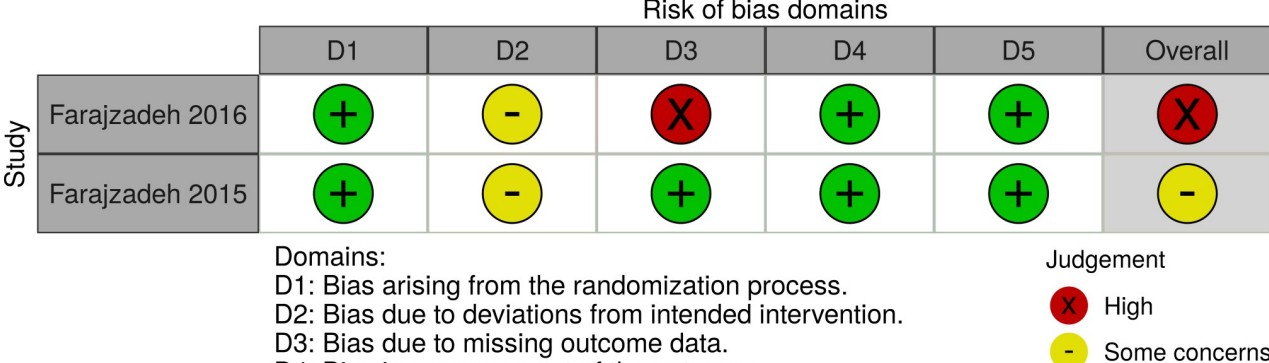

**Fig 2. Risk of Bias assessment of randomised controlled trials.** The Revised Cochrane risk-of-bias tool for randomised trials (RoB2) was applied for the evaluation.

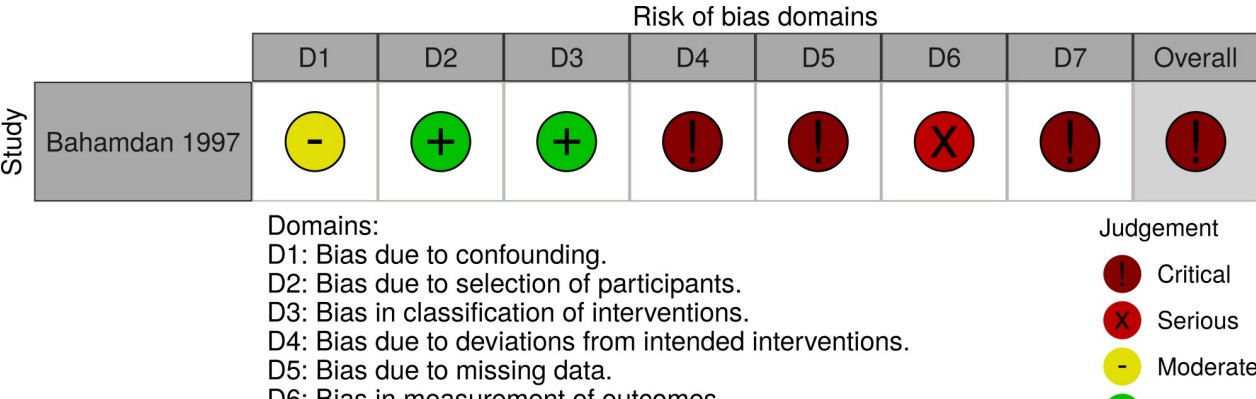

**Fig 3. Risk of Bias assessment of a non-randomised study in humans.** The Risk Of Bias In Non-randomised Studies–of Interventions (ROBINS-I) assessment tool was applied for the evaluation.

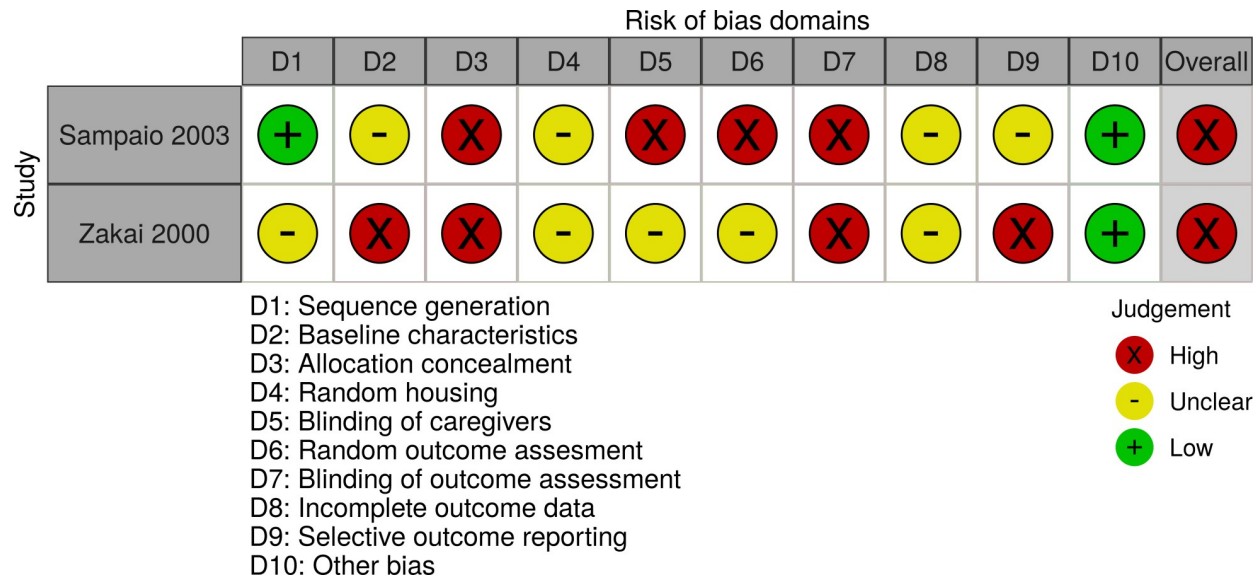

**Fig 4. Risk of Bias assessment of animal trials.** SYRCLE's risk-of-bias tool for animal studies was applied for the evaluation.

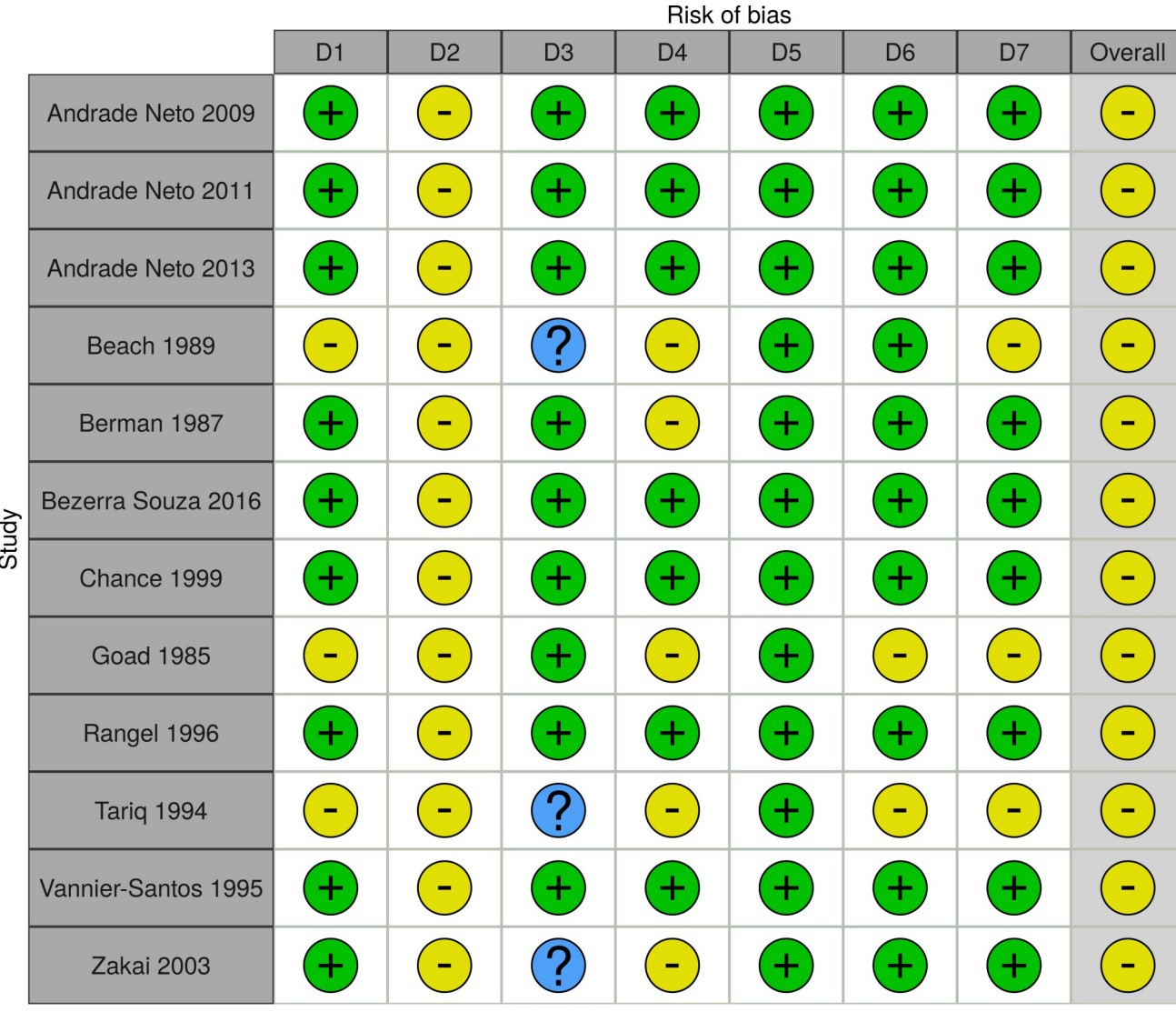

**Fig 5. Risk of Bias assessment of *in vitro* studies.** The risk-of-bias tool to address *in vitro* studies developed by the United States national toxicology program was applied for the evaluation.

The only well-designed randomised controlled trial of Farajzadeh et al. that compared the treatment efficacy of oral terbinafine versus intramuscular meglumine antimoniate showed a non-significant lower cure rate for terbinafine (38% vs 53% of treated patients) [25].

Farajzadeh [26] and Bahmdans [27] clinical trials with terbinafine reported cure rates of 14% and 15% respectively, but the findings of these studies should be interpreted with caution due to high rates of loss to follow up. Two animal studies lacked allocation concealment and did not blind outcome assessment and therefore should be interpreted with caution [28, 29].

**Table 3. Overview of clinical and *in vitro* *Leishmania* species specific results of terbinafine in cutaneous leishmaniasis.**

| *Leishmania* Complex | *Leishmania* Species | Growth inhibition in promastigotes at 27μM | Effective doses in promastigotes | Effective doses in amastigotes | Cure rate in clinical study [a] |
|---|---|---|---|---|---|
| *major* | *major* | 52–90% | IC50: 6μM | ED50: 31μM | NA |
| *tropica* | *tropica* | 92–93% | MIC: 1373μM | NA | 38% |
| | *aethiopica* | 90% | NA | NA | NA |
| *mexicana* | *mexicana* | 26% | no inhibition / MIC: 15–34μM | NA | NA |
| | *amazonensis* | 74% | IC50: 4–9μM / MIC: 1μM | IC50: 23μM / MIC: 0,001μM [b] | NA |
| *braziliensis* | *braziliensis* | 72% | MIC: 1–5μM | NA | NA |
| *guyanensis* | *guyanensis* | 49% | NA | NA | NA |
| | *panamanensis* | 41% | NA | NA | NA |

IC50: Half IC50: maximal inhibitory concentration, ED50: Median effective dose, MIC: Minimum inhibitory concentration, NA: Not Applicable

[a] Defined as decrease in induration size > 75% of lesions at last available at follow up and calculated according to intention to treat analysis

[b] combined with 0,001μM Ketoconazole

The *in vitro* studies showed that terbinafine, butenafine, and naftifine eliminated amastigotes at concentrations between 23 and 45μM, that is approximately five times higher than the terbinafine levels achieved in the skin during terbinafine treatment [46–48]. Therefore, we conclude that allylamines are not promising for CL and MCL treatment.

**Table 4. Results of *in vitro* and clinical studies on the combination of terbinafine with other treatment in cutaneous and mucocutaneous leishmaniasis.**

| Study | *Leishmania* species | Target | Combined therapy | Result |
|---|---|---|---|---|
| Andrade-Neto 2013[30] | *L. amazonensis* | promastigote | LBqT01 | synergistic effect[a] |
| Andrade-Neto 2013 | *L. amazonensis* | promastigote | imipramine | additive effect[b] |
| Andrade-Neto 2013 | *L. amazonensis* | amastigote | LBqT01 | no significant effect |
| Andrade-Neto 2013 | *L. amazonensis* | amastigote | imipramine | no significant effect |
| Vannier Santos 1995 [39] | *L. amazonensis* | amastigote | ketoconazole | synergistic effect[c] |
| Rangel 1996 [37] | *L. braziliensis* | promastigote | ketoconazole | synergistic effect[d] |
| Rangel 1996 | *L. braziliensis* | promastigote | D0870 | synergistic effect[d] |
| Vellin 2005 [45] | *L. infantum* | MCL | itraconazole | complete epithelialization |
| Mawenzi 2018 [44] | unknown | CL | crotamiton + sulfur | complete epithelialization |
| Farajzadeh 2015 [25] | *L. tropica* | CL | cryotherapy | no significant effect |
| Farajzadeh 2016 [26] | *L. tropica* | CL | meglumine antimoniate | no significant effect |

[a] Synergism defined as fractional inhibitory concentration index (FICI) ≤ 0,5

[b] Additive effect defined as: 0,5 < FICI < 4

[c] Synergism defined as total fractional inhibition higher than expected from adding up the fractional inhibition of each individual drug

[d] Synergism defined as 300-fold reduction of the Minimum Inhibitory Concentration of ketoconazole with 1μM terbinafine.

Farajzadeh recommends terbinafine as an alternative to meglumine antimoniate in the case of allergy or resistance [25]. Although the work reports on hazard ratios and time to healing, it does not mention the complete cure rates in the abstract and conclusion sections. The cure rate of 38% was not significantly lower than the 53% cure rate with meglumine antimoniate, and we consider it too low to propose it as a new alternative treatment. The lack of significance of the lower cure rate of terbinafine compared to meglumine antimoniate could be explained by a low effectivity of the latter.

Whilst this review shows that there is no evidence for efficacy of terbinafine in the treatment of CL and MCL, it is highly effective in the treatment of mycotic skin disease. The difference may be due to the high sensitivity of skin fungus to terbinafine compared to *Leishmania* amastigotes. Terbinafine eliminates skin fungus *in vitro* at a mean concentration of 0,014µM [49], thus is approximately 2500 times more effective than the elimination of *Leishmania* amastigotes.

Promastigote cultures are relatively easy and cheap to maintain but are not very reliable as predictors of in vivo effectivity as they represent the infective mosquito stage of the parasite whilst human infection is sustained by intracellular amastigotes [50, 51]. Therefore, the results of the *in vitro* study of Beach et al. that indicates effective concentrations of 1–34µM of terbinafine in *L. braziliensis* and *L. amazonensis* promastigotes should be interpreted with caution. Results of promastigote studies must be confirmed in amastigote studies.

Although triazole monotherapy does not seem effective as treatment of CL patients, results from *in vitro* studies indicate terbinafine combined with triazole drugs may be effective through a synergistic effect. Terbinafine combined with ketoconazole eliminated *L. amazonensis* amastigotes at levels of 0,001µM of both drugs. Terbinafine would reach those levels with an oral dose of 250mg but the best combination with a triazole drug still has to be defined [46]. Triazole drugs like ketoconazole and fluconazole are inhibitors of the enzymes CYP 2C9 and CYP 3A4, involved in terbinafine metabolism, and may cause significant rise in terbinafine plasma concentrations. Secondary effects of terbinafine combined with triazoles have not been studied extensively and would require large clinical studies before implementation [52, 53].

## Conclusion

Based on a systematic review of available literature we conclude that there is no evidence for the efficacy of allylamine monotherapy against CL and MCL. Further trials of allylamines as a treatment for CL and MCL should be carefully considered as the outcomes of an adequately designed trial were disappointing and *in vitro* studies indicate minimal effective concentrations that are not achieved in the skin during standard doses of 250-1000mg oral terbinafine/day. However, the *in vitro* synergistic effects of allylamines combined with triazole drugs against amastigotes, warrant more investigation starting with high quality animal studies to define optimal doses and safety profiles and followed by well-designed trials in humans in case of positive findings.

## Supporting information

**S1 Table. Characteristics of case reports.**
(DOCX)

**S1 File. Full electronic search.**
(DOCX)

**S2 File. PRISMA 2009 checklist.**
(DOCX)

## Author Contributions

**Conceptualization:** Jacob M. Bezemer, Jacob van der Ende, Jacqueline Limpens, Henry J. C. de Vries, Henk D. F. H. Schallig.

**Formal analysis:** Jacob M. Bezemer, Jacob van der Ende, Jacqueline Limpens.

**Investigation:** Jacob M. Bezemer, Jacob van der Ende, Jacqueline Limpens.

**Methodology:** Jacob M. Bezemer, Jacob van der Ende, Jacqueline Limpens.

**Project administration:** Jacob M. Bezemer.

**Resources:** Henry J. C. de Vries, Henk D. F. H. Schallig.

**Software:** Jacqueline Limpens.

**Supervision:** Henry J. C. de Vries, Henk D. F. H. Schallig.

**Validation:** Jacqueline Limpens, Henk D. F. H. Schallig.

**Writing – original draft:** Jacob M. Bezemer.

**Writing – review & editing:** Jacob van der Ende, Jacqueline Limpens, Henry J. C. de Vries, Henk D. F. H. Schallig.

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
