## [Decision Letter · Decision Letter 0]

16 Dec 2020

PONE-D-20-23096

Safety and efficacy of allylamines in the treatment of tegumentary leishmaniasis: a systematic review

PLOS ONE

Dear Dr. Bezemer,

Thank you for submitting your manuscript to PLOS ONE. After careful consideration, we feel that it has merit but does not fully meet PLOS ONE’s publication criteria as it currently stands. Therefore, we invite you to submit a revised version of the manuscript that addresses the points raised during the review process.

We look forward to receiving your revised manuscript.

Kind regards,

Kristien Verdonck

Academic Editor

PLOS ONE

Journal Requirements:

"JB received a monthly volunteer allowance from Latin Link Nederland http://www.latinlink-nederland.nl/

Latin Link had no role in study design, data collection and analysis, decision to publish, preparation of the manuscript"

We note that you received funding from a commercial source: Latin Link Nederland

Reviewers' comments:

Reviewer's Responses to Questions

**Comments to the Author**

1. Is the manuscript technically sound, and do the data support the conclusions?

Reviewer #1: Partly

Reviewer #2: Yes

Reviewer #3: Partly

2. Has the statistical analysis been performed appropriately and rigorously? 

Reviewer #1: Yes

Reviewer #2: Yes

Reviewer #3: Yes

3. Have the authors made all data underlying the findings in their manuscript fully available?

Reviewer #1: Yes

Reviewer #2: Yes

Reviewer #3: Yes

4. Is the manuscript presented in an intelligible fashion and written in standard English?

Reviewer #1: Yes

Reviewer #2: Yes

Reviewer #3: No

5. Review Comments to the Author

Reviewer #1: The methodology of the study seems sound, and the prisma guidelines have been followed. However, I feel the written manuscript needs significant editing both in content and form before it is suitable for publication.

General

-Editing is needed (see pdf for more examples)

• Numbers spelled out or in number (e.g. line 38 ; one/2)

• Abbreviations not used consistently

• in vitro is sometimes not in italics

• Tables cannot be viewed properly (last columns have fallen off page), please show in 'landscape orientiation'

• Punctuation needs to be checked, sometimes missing or inconsistent, left out wrongly, sometimes overused

-It seems the authors are quite focused on New World leishmaniasis, this should be adjusted if they want to talk about cutaneous and mucocutaneous leishmaniasis in general

• the term tegumentary CL is generally used more for the America's and is not common for Old World CL. But the study is also about L. tropica. Perhaps better to talk about cutaneous and mucocutaneous leishmaniasis?

• Not all leishmania lesions are necessarily ulcers, in many parts of the world, ulcers are not the dominant presentation

Abstract

-Make abstract more general, instead of mainly New World CL. Also now it seems almost all mucosal disease is lethal, which is misleading.

-Not all current TL treatment have severe side-effects. What about local treatments such as cryotherapy or miltefosine?

Introduction

-Rationale of why a review on this topic is needed should be more clearly explained. Why allylamines and not any other drug? Now the main reason why allylamines are proposed is safety. Its oral form, or low cost (?), availability(?) could be emphasized more

-In what capacity do you think allylamines could be used? Once this is defined the introduction can be made more targeted. Now it is explained as an alternative for all CL drugs, which are heaped together as bad and having severe side-effects. But there are plenty of promising treatment initiatives for CL treatment ongoing, thermotherapy, miltefosine.

• If used as an alternative for systemic pentavalent antimonials (where the major concern is safety) then lesions should also be severe enough to warrant systemic treatment.

• If intended for more simple lesions, describing the disadvantages of systemic treatment doesn’t make sense

• If you want to keep all lesions please explain why other promising safe treatment modalities are not an option

Methods

-Most studies are not on animals or humans. Any way to assess bias for in vitro studies?

Results

-Please also discuss studies with higher risk of bias, now only the 2015 trial is discussed. Just because they have bias does not mean they should be disregarded altogether, it just means they should be interpreted with caution.

-I would like to see a bit more info on the case reports. Did these patients fail to respond to other treatments? Why were they started on allylamines otherwise?

-I would prefer one table with all evidence on effectiveness and one table on characteristics of all studies, separated per study type. The high number of tables make it hard to get a clear overview.

-Please reconsider structure of results, results seem fragmented, some repetition (e.g. MCL/VL case)

-Please ensure tables are fully legible before approving pdf submission!

Discussion

-Do we have enough evidence to say allylamines do not work? It is important to stress that the quality of evidence found was poor, all human studies were only for L. tropica, while leishmaniasis is a very heterogenous disease.

-The discussion needs more structure.

-Discussion can be shortened significantly. Now some parts seem more suited for introduction, others for methods, results. There is no need to refer to tables any more in discussion, this is more for results Some more detailed suggestions can be found below.

• Line 218-228 seems to indicate the dose given to patients was too low (this paragraph can be made more concise). Explain whether it is not an option to increase dosage to reach required levels.

• Line 240-245 explain that based on the clinical papers, terbinafine does not seem promising. I think one paragraph summarizing all evidence and stating that the allylamines are not promising is clearer; this should be the first paragraph of the discussion.

• Line 246-248: This is more for introduction, it disrupts the flow of the discussion. Better to leave out details about mechanism of action. "Although triazole monotherapy does not seem effective as for treatment of CL patients, results from in vitro studies indicate terbinafine combined with triazole drugs may be effective through a synergistic effect".

• Line 254-260: this paragraph can be shortened and combined with the previous

Reviewer #2: 1. Summary of the research and overall impression

a. The authors delivered a well written systematic review that assessed the evidence available on the efficacy and safety of allylamine treatment for tegumentary leishmaniasis. The search strategy was extensive and inclusive, the methodology accurate and well described, and the conclusions clearly formulated. As a note, please be aware that the tables were not fully available (cut-off on the right side of the text) for reviewing, which might have influenced the comments made by the reviewer.

2. Evidence and examples

a. Major issues

i. Line 229 and 261: ‘this review shows that terbinafine is not effective’ versus line 244: ‘lack of evidence for the efficacy of terbinafine’. The authors base this conclusion primarily on one well-designed human trial, together with (indirect) arguments coming from in vitro studies (although this impression might be linked to incomplete availability of the data from the tables). From the text, it seems more correct to state that there is no proof of efficacy, rather than state that there is proof of inefficacy.

b. Minor issues

i. Abstract

1. Line 71: ‘Furthermore, poor treatment responses exist for TL drugs’. Please specify: is this so in general/for the majority of patients? Or specifically so for pentavalent antimonials,…? How does this sentence add to line 66 ‘… each species responds differently to treatment’?

2. Line 78: add space between ‘death[8]’

3. Line 84: ‘All original human, animal and in vitro studies, including studies comparing effects of allylamines with placebo or alternative TL treatments, were eligible for inclusion’. This sentence seems strangely formulated. Alternative suggestion: ‘all original human, animal and in vitro studies concerning allylamines and leishmaniasis were eligible for inclusion. Comparators – if any - included both placebo or alternative TL treatments’ ?

ii. Methods

1. Line 92: add period at the end of the line.

2. Line 97: ‘no language, date or other restriction were applied’. Replace: either ‘other restrictionS were applied’ or ‘other restriction WAS applied’.

3. Line 125: ‘presentation’. Add ‘drug’ for clarity (as was done for human studies in line 120).

iii. Results

1. Line 175: add space between ‘2015[21]’.

2. In the tables with Characteristics of trials (Table 1-4) it is not registered systematically whether or not the treatment was allylamine monotherapy or combination treatment. For human trials and case reports, a combination column is present, but this is not the case for mice studies (Table 2) or in vitro studies. I would suggest the authors to include this column for all tables on characteristics. At present it seems from the table 3 and 4 that in vitro studies all studied allylamine monotherapy, while it only becomes clear in line 194 that most tested combination therapies. [Since Tables 1-4 are not fully visible in the manuscript provided, it is possible that this column is indeed present in all tables, in which case the authors can ignore this comment.]

3. Line 196: add space to ‘L.braziliensis’. Similar mistake on line 198, 219, 237, 238, 244, 254, 264

iv. Discussion

1. Line 218: sentence is difficult to understand, try to rephrase? E.g. ‘A low sensitivity of amastigotes to the clinically achievable levels of terbinafine…’?

2. Line 256: add space to ‘defined[41]’

Reviewer #3: 

The aim of this review article is to assess the safety and efficacy of allylamines in the treatment of tegumentary leishmaniasis. In this search, 22 publications were included, PRISMA statement for reporting was used and also risk of bias in human and animal studies was assessed with the Cochrane and SYRCLE´s tools, respectively.

Overall, the study is interesting and novel; however, the following concerns need to be corrected:

Abstract:

- The conclusion section is brief; please explain more about the message you want to present.

- You shouldn’t sort out the abstract, please check the instruction of the journal.

Introduction:

- Line 61: please delete “Background”.

Methods:

- Line 130: “Risk of bias”, please explain more about this section.

Results:

- Tables 1,2,3,4,8: I couldn't see some columns of these tables, please make the tables in the form of landscape or smaller.

- Line 153: After “One animal study reported the efficacy of terbinafine on L. amazonensis and the other on L. major” please mention the references.

- Line 155: After “allylamines in amastigote models” mention the references.

- Line 183: In the study of Farajzadeh et al., 2015, I found that oral terbinafine may have approximately the same efficacy as glucantime. So please explain more about the finding and conclusion of the study in both discussion and result sections.

- Line 191: After “compared to the median cytotoxic concentration of 98μ” please mention the references.

- Line 193: After “were more than 110 μM” please mention the references.

- Line 199: After “resulting in a minimally inhibitory concentration of 0,001μM.” please mention the references.

Discussion:

- Line 228: As I said, in the study of Farajzadeh et al., 2015; oral terbinafine demonstrates the same efficacy as glucantime. it seems that in some cases, oral terbinafine could be used for treatment of cutaneous leishmaniasis. So you should explain more about the finding of that study in the discussion.

Conclusion:

- The conclusion section is brief; explain more about the message you want to deliver.

6. PLOS authors have the option to publish the peer review history of their article (what does this mean?). If published, this will include your full peer review and any attached files.

Reviewer #1: No

Reviewer #2: No

Reviewer #3: No

---

## [Author Response · Author response to Decision Letter 0]

4 Feb 2021

Response to Reviewers

Reviewer #1: The methodology of the study seems sound, and the prisma guidelines have been followed. However, I feel the written manuscript needs significant editing both in content and form before it is suitable for publication.

General

-Editing is needed (see pdf for more examples)

Response: Thank you very much for the revision. The comments in the PDF were applied to the manuscript as applicable. We want to explain that the outcomes are described at the end of the introduction in compliance with the PRISMA checklist. In the discussion we use the comparison with the effectivity of terbinafine in mycotic disease because it is the only generally accepted application of terbinafine in humans. 

• Numbers spelled out or in number (e.g. line 38; one/2)

Response: Spelling of the numbers was corrected throughout the manuscript.

• Abbreviations not used consistently

Response: The use of abbreviations was corrected throughout the manuscript.

• in vitro is sometimes not in italics

Response: In vitro was written in italics throughout the manuscript.

• Tables cannot be viewed properly (last columns have fallen off page), please show in 'landscape orientiation'

Response: We apologize for the loss of columns during PDF creation and changed the text orientation of the tables that missed columns.

• Punctuation needs to be checked, sometimes missing or inconsistent, left out wrongly, sometimes overused

Response: Punctuation was corrected throughout the manuscript.

-It seems the authors are quite focused on New World leishmaniasis, this should be adjusted if they want to talk about cutaneous and mucocutaneous leishmaniasis in general

Response: Thanks to your comment, the focus of the written text was adjusted to leishmaniasis in general. 

• the term tegumentary CL is generally used more for the America's and is not common for Old World CL. But the study is also about L. tropica. Perhaps better to talk about cutaneous and mucocutaneous leishmaniasis?

Response: Thanks to your comment, the term tegumentary was replaced with cutaneous and mucocutaneous throughout the manuscript, including the title.

• Not all leishmania lesions are necessarily ulcers, in many parts of the world, ulcers are not the dominant presentation

Response: Thanks to your comment, the text was adapted to include the different presentations of cutaneous leishmaniasis. 

Abstract

-Make abstract more general, instead of mainly New World CL. Also, now it seems almost all mucosal disease is lethal, which is misleading.

Response: Thanks to your comment the focus of the text was adapted to leishmaniasis in general.

-Not all current TL treatment have severe side-effects. What about local treatments such as cryotherapy or miltefosine?

Response: Thanks to your comment, the statement about severe side-effects was adapted.

Introduction

-Rationale of why a review on this topic is needed should be more clearly explained. Why allylamines and not any other drug? Now the main reason why allylamines are proposed is safety. Its oral form, or low cost (?), availability(?) could be emphasized more

Response: Thanks to your suggestion the oral form, availability, and cost of terbinafine was mentioned in the introduction. 

-In what capacity do you think allylamines could be used? Once this is defined the introduction can be made more targeted. Now it is explained as an alternative for all CL drugs, which are heaped together as bad and having severe side-effects. But there are plenty of promising treatment initiatives for CL treatment ongoing, thermotherapy, miltefosine.

Response: Allylamines could potentially be an oral or local, safe, easy to administer, and affordable alternative for the current arsenal of leismananial treatment options, many of whom are either expensive (e.g. miltefosine), painful (e.g. intralesional antimony) or bear severe side effects (e.g. systemic antimony or amphothericine-B) 

• If used as an alternative for systemic pentavalent antimonials (where the major concern is safety) then lesions should also be severe enough to warrant systemic treatment.

Response: Our objective is a general systematic review that includes all possible cutaneous and mucocutaneous forms. Since the scarcity of data, we did not confine the review to severe forms that warrant systemic therapy only.

• If intended for more simple lesions, describing the disadvantages of systemic treatment does not make sense.

Response: Please, see the above written comment.

• If you want to keep all lesions please explain why other promising safe treatment modalities are not an option.

Response: Thank you very much for your valuable comment, we added the existence of multiple alternative treatments options in the introduction. The objective of this review is the utility of allylamines for cutaneous and mucocutaneous leishmaniasis. We consider it out of the scope of this paper to analyze the other possible promising safe treatment modalities, as this review compares with the mainstay treatment that’s pentavalent antimonials up to date. 

Methods

-Most studies are not on animals or humans. Any way to assess bias for in vitro studies?

Response: Although no internationally accepted method for assessing bias in in vitro studies exists, thanks to your comment we applied the method developed by the US National Toxicology Program and implemented the figure in the manuscript.

Results

-Please also discuss studies with higher risk of bias, now only the 2015 trial is discussed. Just because they have bias does not mean they should be disregarded altogether; it just means they should be interpreted with caution.

Response: We thank you for your observation and kindly remind you that the results of the studies with high risk of bias can be observed in table 1, which is completely visible now. Additionally, we added a phrase in the methods section in the paragraph `Risk of bias assessment´ stating that studies with a high risk of bias are excluded from the analysis.

We feel that it is not wise to discuss the results of studies with high risk of bias, because it might suggest that those results were reliable. We clarified that in the methods section.

-I would like to see a bit more info on the case reports. Did these patients fail to respond to other treatments? Why were they started on allylamines otherwise?

Response: Thanks to your comment, the reason to start with terbinafine was stated in the S2 table and in the results section. 

-I would prefer one table with all evidence on effectiveness and one table on characteristics of all studies, separated per study type. The high number of tables make it hard to get a clear overview.

Response: Although the heterogeneity of the studies did not permit the aggregation of all tables as proposed, we combined several tables and changed the ROB tables into figures, hoping that it will improve the manuscript reading. 

-Please reconsider structure of results, results seem fragmented, some repetition (e.g. MCL/VL case)

Response: Thanks to your comments, the result section was restructured. 

-Please ensure tables are fully legible before approving pdf submission!

Response: We apologize for the lost columns and adapted the table orientation. 

Discussion

-Do we have enough evidence to say allylamines do not work? It is important to stress that the quality of evidence found was poor, all human studies were only for L. tropica, while leishmaniasis is a very heterogenous disease.

Response: Thanks to your comment the conclusion was adapted. 

-The discussion needs more structure.

Response: Thanks to your comment the discussion was restructured.

-Discussion can be shortened significantly. Now some parts seem more suited for introduction, others for methods, results. There is no need to refer to tables any more in discussion, this is more for results Some more detailed suggestions can be found below.

Response: Thanks to your comment the references to tables were eliminated from the discussion.

• Line 218-228 seems to indicate the dose given to patients was too low (this paragraph can be made more concise). Explain whether it is not an option to increase dosage to reach required levels.

Response: Thanks to your comment the paragraph was adapted.

• Line 240-245 explain that based on the clinical papers, terbinafine does not seem promising. I think one paragraph summarizing all evidence and stating that the allylamines are not promising is clearer; this should be the first paragraph of the discussion.

Response: Thanks to your comment, the first paragraph was adapted.

• Line 246-248: This is more for introduction, it disrupts the flow of the discussion. Better to leave out details about mechanism of action. "Although triazole monotherapy does not seem effective as for treatment of CL patients, results from in vitro studies indicate terbinafine combined with triazole drugs may be effective through a synergistic effect".

Response: We applicated your suggestion.

• Line 254-260: this paragraph can be shortened and combined with the previous

Response: We applicated your suggestion.

Reviewer #2: 1. Summary of the research and overall impression

a. The authors delivered a well written systematic review that assessed the evidence available on the efficacy and safety of allylamine treatment for tegumentary leishmaniasis. The search strategy was extensive and inclusive, the methodology accurate and well described, and the conclusions clearly formulated. As a note, please be aware that the tables were not fully available (cut-off on the right side of the text) for reviewing, which might have influenced the comments made by the reviewer.

2. Evidence and examples

a. Major issues

i. Line 229 and 261: ‘this review shows that terbinafine is not effective’ versus line 244: ‘lack of evidence for the efficacy of terbinafine’. The authors base this conclusion primarily on one well-designed human trial, together with (indirect) arguments coming from in vitro studies (although this impression might be linked to incomplete availability of the data from the tables). From the text, it seems more correct to state that there is no proof of efficacy, rather than state that there is proof of inefficacy. 

Response: Thank you very much for the revision and comments. The suggestion was applied.

b. Minor issues

i. Abstract

1. Line 71: ‘Furthermore, poor treatment responses exist for TL drugs’. Please specify: is this so in general/for the majority of patients? Or specifically so for pentavalent antimonials,…? How does this sentence add to line 66 ‘… each species responds differently to treatment’?

Response: Thanks to your suggestion the sentence was changed.

2. Line 78: add space between ‘death[8]’

Response: Thanks for the comment, it was applied. 

3. Line 84: ‘All original human, animal and in vitro studies, including studies comparing effects of allylamines with placebo or alternative TL treatments, were eligible for inclusion’. This sentence seems strangely formulated. Alternative suggestion: ‘all original human, animal and in vitro studies concerning allylamines and leishmaniasis were eligible for inclusion. Comparators – if any - included both placebo or alternative TL treatments’ ?

Response: Thank you very much for the suggestion, it was applied.

ii. Methods

1. Line 92: add period at the end of the line.

Response: Thanks for the comment, it was applied.

2. Line 97: ‘no language, date or other restriction were applied’. Replace: either ‘other restrictionS were applied’ or ‘other restriction WAS applied’.

Response: Thanks for the comment, it was applied.

3. Line 125: ‘presentation’. Add ‘drug’ for clarity (as was done for human studies in line 120).

Response: Thanks for the comment, it was applied.

iii. Results

1. Line 175: add space between ‘2015[21]’.

Response: Thanks for the comment, it was applied.

2. In the tables with Characteristics of trials (Table 1-4) it is not registered systematically whether or not the treatment was allylamine monotherapy or combination treatment. For human trials and case reports, a combination column is present, but this is not the case for mice studies (Table 2) or in vitro studies. I would suggest the authors to include this column for all tables on characteristics. At present it seems from the table 3 and 4 that in vitro studies all studied allylamine monotherapy, while it only becomes clear in line 194 that most tested combination therapies. [Since Tables 1-4 are not fully visible in the manuscript provided, it is possible that this column is indeed present in all tables, in which case the authors can ignore this comment.]

Response: Thank you very much for the comment. We added the column indicating combination treatments to the tables with characteristics of trials and in vitro studies (Table 1 and 2 after fusion).

3. Line 196: add space to ‘L.braziliensis’. Similar mistake on line 198, 219, 237, 238, 244, 254, 264

Response: Thanks for the comment, it was applied.

iv. Discussion

1. Line 218: sentence is difficult to understand, try to rephrase? E.g. ‘A low sensitivity of amastigotes to the clinically achievable levels of terbinafine…’?

Response: Thanks for the comment, it was applied.

2. Line 256: add space to ‘defined[41]’

Response: Thanks for the comment, it was applied.

Reviewer #3: 

The aim of this review article is to assess the safety and efficacy of allylamines in the treatment of tegumentary leishmaniasis. In this search, 22 publications were included, PRISMA statement for reporting was used and also risk of bias in human and animal studies was assessed with the Cochrane and SYRCLE´s tools, respectively.

Overall, the study is interesting and novel; however, the following concerns need to be corrected:

Abstract:

- The conclusion section is brief; please explain more about the message you want to present.

Response: Thanks to your comment, we extended the conclusion section of the abstract.

- You shouldn’t sort out the abstract, please check the instruction of the journal.

Response: We corrected the headings in the abstract as you suggest. 

Introduction:

- Line 61: please delete “Background”.

Response: Thanks for the comment, it was applied.

Methods:

- Line 130: “Risk of bias”, please explain more about this section.

Response: Thanks to your comment, the Risk of bias section was extended.

Results: 

- Tables 1,2,3,4,8: I couldn't see some columns of these tables, please make the tables in the form of landscape or smaller.

Response: We apologize for the loss of columns during the transformation to PDF. It was corrected.

- Line 153: After “One animal study reported the efficacy of terbinafine on L. amazonensis and the other on L. major” please mention the references.

Response: The phrase was removed from the manuscript in response to other comments. 

- Line 155: After “allylamines in amastigote models” mention the references.

Response: The phrase was removed from the manuscript in response to other comments.

- Line 183: In the study of Farajzadeh et al., 2015, I found that oral terbinafine may have approximately the same efficacy as glucantime. So please explain more about the finding and conclusion of the study in both discussion and result sections.

Response: Thanks to your comment we mention the difference in endpoint between Farajzadeh and our systematic review. 

- Line 191: After “compared to the median cytotoxic concentration of 98μ” please mention the references.

Response: Thanks for the comment, it was applied.

- Line 193: After “were more than 110 μM” please mention the references.

Response: Thanks for the comment, it was applied.

- Line 199: After “resulting in a minimally inhibitory concentration of 0,001μM.” please mention the references.

Response: Thanks for the comment, it was applied.

Discussion:

- Line 228: As I said, in the study of Farajzadeh et al., 2015; oral terbinafine demonstrates the same efficacy as glucantime. it seems that in some cases, oral terbinafine could be used for treatment of cutaneous leishmaniasis. So you should explain more about the finding of that study in the discussion.

Response: Thanks to your comment we explain that the same efficacy was possibly caused by resistance to meglumine antimoniate. 

Conclusion:

- The conclusion section is brief; explain more about the message you want to deliver.

Response: Thanks to your comment we extended the conclusion section.

---

## [Decision Letter · Decision Letter 1]

3 Mar 2021

PONE-D-20-23096R1

Safety and efficacy of allylamines in the treatment of cutaneous and mucocutaneous leishmaniasis: a systematic review

PLOS ONE

Dear Dr. Bezemer,

Thank you for submitting your manuscript to PLOS ONE. After careful consideration, we feel that it has merit but does not fully meet PLOS ONE’s publication criteria as it currently stands. Therefore, we invite you to submit a revised version of the manuscript that addresses the points raised during the review process.

Apart from a few minor issues, I would like to invite you to consider the suggestion to either include the studies at high risk of bias or to make a better case for excluding them (maybe based on study design? see below). 

We look forward to receiving your revised manuscript.

Kind regards,

Kristien Verdonck

Academic Editor

PLOS ONE

Journal Requirements:

Additional Editor Comments:

Abstract and methods: “Studies were excluded from analysis if high risk of bias was perceived during assessment”. How many studies were excluded in this way? This is an unusual approach, especially because you include studies with diverse designs. Unless you have good reasons to do otherwise, I would prefer that you use “study design” at the level of study selection, and report the “risk of bias assessment” as part of the findings (not as an exclusion criterion) and in the discussion. Within the findings of the review, you can then highlight that some studies had high risk of bias (explain the reasons) and that the findings of these studies should be interpreted with caution.Abstract: terbinafine is safe during pregnancy. It is not only about safety during pregnancy? Consider rephrasing.Introduction: “antimonial therapy is painful”. Please revise this sentence; I assume it is only painful when applied in the lesion.Pentavalent antimonials are the most frequently used drugs for new world leishmaniasis. Is that also true for old world leishmaniasis? See for example https://apps.who.int/iris/bitstream/handle/10665/44412/WHO_TRS_949_eng.pdf?sequence=1&isAllowed=y (Box 2, page 69).Methods, study eligibility: check spelling of “species”. The singular is species, not “specie”.

Reviewers' comments:

Reviewer's Responses to Questions

**Comments to the Author**

1. If the authors have adequately addressed your comments raised in a previous round of review and you feel that this manuscript is now acceptable for publication, you may indicate that here to bypass the “Comments to the Author” section, enter your conflict of interest statement in the “Confidential to Editor” section, and submit your "Accept" recommendation.

Reviewer #2: All comments have been addressed

Reviewer #3: All comments have been addressed

2. Is the manuscript technically sound, and do the data support the conclusions?

Reviewer #2: Yes

Reviewer #3: Yes

3. Has the statistical analysis been performed appropriately and rigorously? 

Reviewer #2: Yes

Reviewer #3: Yes

4. Have the authors made all data underlying the findings in their manuscript fully available?

Reviewer #2: Yes

Reviewer #3: Yes

5. Is the manuscript presented in an intelligible fashion and written in standard English?

Reviewer #2: Yes

Reviewer #3: Yes

6. Review Comments to the Author

Reviewer #2: minor suggestions:

- Line 35: remove comma after Leishmania tropica

- Line 51: ‘… are still the most frequently used ones’ ?

- Line 139 + 261: et al. (dot behind al)

- Line 250 : to propose it as (rather than propose it for)?

- Numbers are sometimes written in words, sometimes in numbers; this is not done coherently. Rule of thumb: numbers ten or below should be written out in words; above ten should be written as numbers

Reviewer #3: (No Response)

7. PLOS authors have the option to publish the peer review history of their article (what does this mean?). If published, this will include your full peer review and any attached files.

Reviewer #2: No

Reviewer #3: No

---

## [Author Response · Author response to Decision Letter 1]

19 Mar 2021

Response to Reviewers 

Apart from a few minor issues, I would like to invite you to consider the suggestion to either include the studies at high risk of bias or to make a better case for excluding them (maybe based on study design? see below). 

Response: Thank you very much for the revision. The studies at high risk of bias were not excluded from the systematic review, as explained below, and the phrases “Studies were excluded from analysis if high risk of bias was perceived during assessment” were eliminated from the abstract and methods.

Journal Requirements:

Response: The reference list was checked for retractions with the `scite reference check´ online tool, but none was detected.

Additional Editor Comments:

• Abstract and methods: “Studies were excluded from analysis if high risk of bias was perceived during assessment”. How many studies were excluded in this way? This is an unusual approach, especially because you include studies with diverse designs. Unless you have good reasons to do otherwise, I would prefer that you use “study design” at the level of study selection, and report the “risk of bias assessment” as part of the findings (not as an exclusion criterion) and in the discussion. Within the findings of the review, you can then highlight that some studies had high risk of bias (explain the reasons) and that the findings of these studies should be interpreted with caution.

Response: Thank you for the important comment. We recognize that the phrases about exclusion of studies at high risk of bias in the abstract and methods generated confusion as it was not our intention to exclude them from the entire review as can be seen in Table 1. Therefore, the two phrases were eliminated (line 31 and line 126 in the Revised Manuscript with track changes) and an additional comment was written in the discussion (line 243-246 in the Revised Manuscript with track changes). 

Abstract: terbinafine is safe during pregnancy. It is not only about safety during pregnancy? Consider rephrasing.

Response: Thanks to your comment, the phrase was adapted to: `Allylamine drugs, like terbinafine, are safe, including during pregnancy. ´

• Introduction: “antimonial therapy is painful”. Please revise this sentence; I assume it is only painful when applied in the lesion.

Response: Thanks for your comment, but we prefer to maintain the sentence as antimonial therapy is either applied intra-lesional, intra-muscular, or intra-venously. We consider all three methods to be painful as they require injections. 

• Pentavalent antimonials are the most frequently used drugs for new world leishmaniasis. Is that also true for old world leishmaniasis? See for example https://apps.who.int/iris/bitstream/handle/10665/44412/WHO_TRS_949_eng.pdf?sequence=1&isAllowed=y (Box 2, page 69).

Response: Thanks to your comment we adapted the statement for old world leishmaniasis to: ` are still the most frequently used for American CL and MCL [5] and frequently used for old world leishmaniasis [6].´(line 53 in the Revised Manuscript with track changes)

• Methods, study eligibility: check spelling of “species”. The singular is species, not “specie”.

Response: Thanks to your comment, the spelling of species was corrected

6. Review Comments to the Author

Reviewer #2: minor suggestions:

- Line 35: remove comma after Leishmania tropica

Response: Thanks to your comment, we removed the comma.

- Line 51: ‘… are still the most frequently used ones’ ?

Response: Thanks to your comment we adapted the statement for old world leishmaniasis. 

- Line 139 + 261: et al. (dot behind al)

Response: Thanks to your comment, we added the dots behind al. 

- Line 250 : to propose it as (rather than propose it for)?

Response: Thanks to your comment, the phrase was adapted.

- Numbers are sometimes written in words, sometimes in numbers; this is not done coherently. Rule of thumb: numbers ten or below should be written out in words; above ten should be written as numbers

Response: Thanks for your comment. We corrected a number 20 and a number 12 that were written out in words in the first paragraph of the introduction and of the discussion.

---

## [Editor Report · Decision Letter 2]

23 Mar 2021

Safety and efficacy of allylamines in the treatment of cutaneous and mucocutaneous leishmaniasis: a systematic review

PONE-D-20-23096R2

Dear Dr. Bezemer,

We’re pleased to inform you that your manuscript has been judged scientifically suitable for publication and will be formally accepted for publication once it meets all outstanding technical requirements.

Kind regards,

Kristien Verdonck

Academic Editor

PLOS ONE
---

## [Editor Report · Acceptance letter]

29 Mar 2021

PONE-D-20-23096R2 

Safety and efficacy of allylamines in the treatment of cutaneous and mucocutaneous leishmaniasis: a systematic review 

Dear Dr. Bezemer:

I'm pleased to inform you that your manuscript has been deemed suitable for publication in PLOS ONE. Congratulations! Your manuscript is now with our production department. 

Kind regards, 

on behalf of

Dr. Kristien Verdonck 

Academic Editor

PLOS ONE